# Exercise Programming Modelling a Standard of Care Approach Improves Physical Health and Patient-Reported Outcomes in Individuals Living with Breast Cancer: A Pilot Study

Stephanie J. Kendall [1,2], Stefan Heinze [2,3], Chris Blanchard [4], Joy C. Chiekwe [1], Jennifer Melvin [5], Nicole Culos-Reed [6], Margaret L. McNeely [7,8], Melanie R. Keats [1,2,5] and Scott A. Grandy [1,2,5,*]

1   School of Health and Human Performance, Dalhousie University, Halifax, NS B3H 4R2, Canada; steph.kendall@dal.ca (S.J.K.); joy.chiekwe@dal.ca (J.C.C.); melanie.keats@dal.ca (M.R.K.)
2   Beatrice Hunter Cancer Research Institute, Halifax, NS B3H 4R2, Canada; stefan.heinze.milne@dal.ca
3   Cancer Care Program, Nova Scotia Health Authority, Halifax, NS B3H 2Y9, Canada
4   Department of Medicine, Dalhousie University, Halifax, NS B3H 4R2, Canada; chris.blanchard@nshealth.ca
5   Department of Medicine, Division of Medical Oncology, Nova Scotia Health, Halifax, NS B3H 2Y9, Canada; jene.melvin@nshealth.ca
6   Department of Kinesiology, University of Calgary, Calgary, AB T2N 1N4, Canada; nculosre@ucalgary.ca
7   Physical Therapy, University of Alberta, Edmonton, AB T6G 2G4, Canada; mmcneely@ualberta.ca
8   Supportive Care, Cancer Care Alberta, Edmonton, AB T5J 3E4, Canada
*   Correspondence: scott.grandy@dal.ca

**Abstract:** Controlled study designs widely report that exercise improves the health of individuals living with breast cancer. Still, many individuals living with breast cancer are not active enough to experience the benefits of exercise. The Activating Cancer Communities through an Exercise Strategy for Survivors study was developed to reach more individuals living with cancer. This report describes the effects of a 12-week individualized exercise program that models a standard-of-care approach on body composition, physical fitness, and patient-reported outcomes in individuals living with breast cancer. Individuals living with breast cancer were recruited for the study and completed an exercise program twice weekly overseen by a Clinical Exercise Physiologist. A total of 43 participants completed the exercise intervention, and 36 withdrew from the study. All participants had significantly improved aerobic fitness, waist circumference, hip circumference, lower body endurance, physical activity behaviour, health-related quality of life, emotional status, and fatigue levels after completing the program. Flexibility, balance, and sleep scores did not change. The results from the 12-week individualized exercise program largely align with the results from more controlled study designs. These results support future initiatives integrating exercise therapy into the standard of care for individuals living with breast cancer.

**Keywords:** exercise; breast cancer; standard of care

## 1. Introduction

Numerous studies indicate that exercise interventions improve a plethora of health-related outcomes for individuals living with breast cancer (IBC), including aerobic fitness [1–4], quality of life [2,4–6], flexibility [6], fatigue [2,5,6], systolic blood pressure [4], body composition [6], depression [1,3], and anxiety [3]. Exercise is also indicated to improve health outcomes in other chronic conditions including diabetes, respiratory disease, and cardiovascular diseases [7]. Unfortunately, many IBC often are not sufficiently active to experience the benefits of exercise [8]. The reasons are likely multifactorial but may include cancer burden and a lack of access to appropriate programming and cancer exercise specialists, knowledge of benefits, motivation, and referral by oncology care providers [9–11]. Moreover, resource accessibility, physician knowledge, and expertise surrounding cancer-specific exercise are often limiting factors in a clinical setting [12]. Given the wealth of

positive evidence, there has been a widespread call to make exercise a standardized part of the cancer care plan [13]. Translating clinical research findings into standard practice could be further facilitated by an implementation–effectiveness approach to research, which utilizes usual care or real-world conditions [14,15]. Results from implementation–effectiveness trials are more transferable into non-research environments as they have high ecological validity, which can inform how to best implement exercise programming for IBC [15,16].

Given the relative lack of cancer-specific exercise programming in Nova Scotia, we aimed to improve access through the development of a network of clinical and community partnerships, provision of cancer-specific exercise education and training (www. thrivehealthservices.com), building clinician and self-referral pathways, participant screening and triage to qualified fitness professionals, and appropriate exercise programming. To this end, we created the Activating Cancer Communities through an Exercise Strategy for Survivors (ACCESS) pilot study [17]. The overall goal was to evaluate implementation strategies for exercise programming embedded in cancer care and to examine the effectiveness of individualized exercise programming in improving health outcomes. Here, we report on the implementation and effectiveness of an exercise program designed, delivered and/or supervised by a Clinical Exercise Physiologist (CEP) on multiple health outcomes for the IBC subset of our ACCESS study. By offering individualized, evidence-based exercise programming, the ultimate goal is to ameliorate the impact of a cancer diagnosis on long-term physical and psychological health in cancer patients and survivors.

This study's primary purpose was to examine how to implement an individualized exercise intervention delivered to IBC and the effects of the exercise intervention on IBC regarding physical health and patient-reported outcomes in a real-world setting.

## 2. Materials and Methods

### 2.1. Study Design and Procedures

ACCESS is a Type II hybrid implementation–effectiveness trial [18] used to evaluate the effectiveness of the delivery of a multisite exercise intervention. Data collection for this IBC subset occurred between 28 September 2018 and 12 December 2019. The study received research ethics board approval from Nova Scotia Health (Halifax, NS, Canada) and was registered (ClinicalTrial.gov Identifier: NCT03599843).

Individuals who consented to study participation underwent a baseline assessment, including completing a pre-intervention questionnaire and fitness assessment. Pending the completion of the baseline assessment, the ACCESS CEP triaged the participant to the appropriate exercise facility. While exercise has been deemed safe for most individuals with a cancer diagnosis, those prescribing exercise must have had the training and experience to implement evidence-based recommendations. Thus, all referrals to ACCESS and consenting participants were triaged through a CEP [19–21]. High-risk participants were required to complete the exercise intervention at the Physical Activity and Cancer (PAC Lab) at Queen Elizabeth II (QEII) Health Sciences Center (Halifax, NS, Canada). Moderate-to-low-risk individuals were given the option of completing the exercise program in hospital or at one of two community-based sites. The community-based sites were supported by the ACCESS CEP and served those participants cleared to exercise under a "Cancer and Exercise" trained fitness professional, but without medical needs that would require direct CEP supervision. Community-based sites were selected based on population density, participant interest, and facility support. Participants were considered high-risk participants if they (1) had a previous cardiac event (e.g., myocardial infarction, stroke); (2) were currently receiving a known cardiotoxic agent (e.g., anthracyclines); and (3) had known bone metastases or advanced stage disease. After completing baseline screening, triage, and assessments, participants received a tailored, 12-week, twice-weekly exercise program. Baseline measures were repeated after the intervention.

### 2.2. Participants

Participants were recruited from the QEII Health Sciences Center cancer clinics and the surrounding community. The study team attended several clinician ground rounds, and nursing huddles, posted a study description in the local health authority magazine, and attended a Nova Scotia cancer care professional development day to describe the study and promote clinician buy-in. Additionally, posters were placed in the cancer clinic waiting and examination rooms. The study was broadcast on local radio stations and the news to recruit IBC directly from the community. Participants were deemed eligible if they (1) had a breast cancer diagnosis; (2) were ≥18 years of age; (3) could participate in at least mild levels of physical activity; (4) were pre-treatment, receiving active treatment, had received a cancer diagnosis within the past 5 years, or had cancer-related side-effects; (5) were able and willing to attend a twice-weekly exercise program; (6) could provide written consent in English; and (7) had medical (i.e., family physician, treating oncologist, oncology nurse, or other qualified health care or CEP) approval to participate. All participants signed an informed consent form before participating in any study-related procedures.

### 2.3. Exercise Intervention

Exercise programs were designed by CEPs and were offered in hospital and community-based settings (i.e., public recreation centers). Programming was delivered by either a CEP (high-risk) or qualified exercise professionals (community-based). All exercise professionals completed cancer and exercise training for fitness professionals (Thrive Cancer and Exercise Training) [22]. At community-based sites, study protocol adherence was monitored by checking in with instructors monthly.

Small group training sessions (8–12 participants) were personalized based on the individual's current and previous health, physical fitness, and lifestyle. Exercise programming was tailored to meet the participant's unique medical needs and daily energy and fatigue levels. Thus, the exercise programming varied from day to day. Each session lasted 45–60 min, including a warm-up and cooldown. Exercise sessions occurred twice weekly and lasted 12 weeks in duration. All participants completed a variety of aerobic, resistance, balance, and flexibility exercises delivered in a circuit-type setting (examples are given in Figure 1).

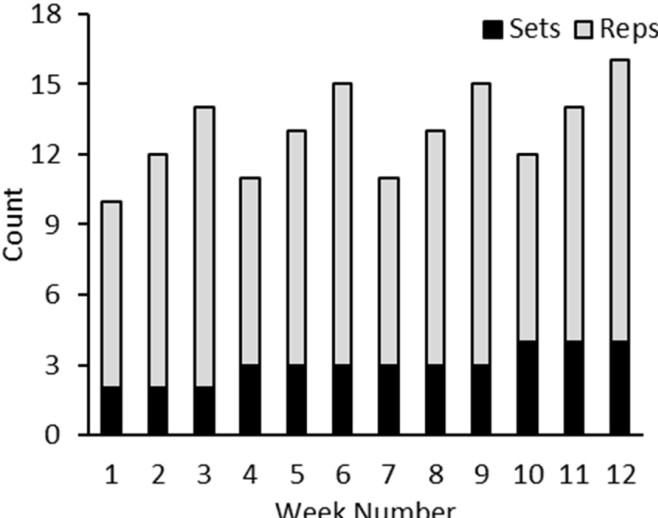

**Figure 1.** Exercise program intensity progression over 12-week exercise program. Examples of exercises include wall ball squat, wall pushup, alternating overhead press, leg extension, tricep extension, plank, tandem stance, clock balance, deadbugs, and walking.

*2.4. Measurements*

Body composition, physical measures, and patient-reported outcomes were assessed at baseline and post-intervention. Data for age, time since diagnosis, treatment type (e.g., surgery, radiation, chemotherapy, hormone treatment), time since starting treatment, and height were collected at baseline to describe the population.

### 2.4.1. Body Composition

Weight, body mass index, blood pressure, resting heart rate, waist circumference, and hip circumference were measured using standard protocols [23].

### 2.4.2. Physical Fitness Measures

Aerobic fitness was estimated using the six-minute walk test (6-MWT) [24]. In brief, participants walked around a straight 100-foot course (i.e., gym or in a hospital hallway), and the total distance completed in six minutes was recorded. The score was recorded to the nearest half lap and converted to distance in metres. Upper body strength was assessed using grip strength [23]. Participants gripped a hand grip dynamometer and completed the test twice on each hand. The average value was calculated for each hand. Maximum scores were recorded to the nearest kilogram. Lower body muscular endurance was assessed using the 30-s chair sit-to-stand test [25]. Participants rose from a chair and sat down as many times as possible in 30 s. Whole repetition counts (seated, standing, and return to seating) were recorded.

Balance was assessed using the one-leg stance test [23]. Participants completed the test on each foot with their eyes open and repeated the assessment with their eyes closed. The test was terminated when the participant could not maintain a sturdy position or completed 45 s. Flexibility was assessed using the seated sit and reach test [25]. Participants sat in a chair with one leg bent and the other extended, leaning as far forward as possible at their hips to reach their toes. The distance between the toes and fingers was recorded.

Shoulder mobility was assessed using the back scratch test [25]. Participants reached one hand behind their head and the other behind their back to bring both hands as close together as possible. The distance between the two middle fingers was recorded to the nearest half-centimetre.

### 2.4.3. Patient-Reported Outcomes

Health-related quality of life (HRQOL) was assessed using the Functional Assessment of Cancer Therapy-General (FACT-G) [26]. The FACT-G is a 27-item, validated questionnaire for assessing the HRQOL of cancer patients. The scale assesses physical, functional, social, and emotional wellbeing and provides subscales for each domain. A lower score indicates poorer HRQOL. The 13-item fatigue subscale, administered in conjunction with the FACT-G, titled the Functional Assessment of Chronic Illness Therapy-Fatigue (FACIT-F), was also administered [27]. The fatigue subscale is a validated questionnaire for assessing fatigue in individuals with cancer. The scale assesses fatigue experience and fatigue impact. A higher score indicates less significant fatigue. The trial outcome index (TOI) was reported, which includes the combined FACT-G and FACIT-F scores.

Physical activity behaviour was assessed using the Godin Leisure Time Exercise Questionnaire (GLTEQ) [28]. The GLTEQ is a three-item questionnaire for assessing physical activity frequency and duration per week and is validated for individuals with cancer [29]. The questionnaire distinguishes between mild (3 metabolic equivalents of task (METS)), moderate (5 METS), and strenuous (9 METS) physical activity, and a higher score indicates greater physical activity levels. The GLTEQ is totalled to a leisure score index (LSI).

Sleep quality was assessed using the Pittsburgh Sleep Quality Index (PSQI) [30]. The PSQI is a 19-item questionnaire for assessing sleep and is validated for individuals with cancer [31]. The PSQI assesses daytime disturbances, sleep quality, latency, duration, efficiency, and medications as subscales. A total score $\geq$ five indicates poor sleep quality [30].

Emotional states of depression, anxiety, and stress were assessed using the Depression Anxiety Stress Scales 42-item (DASS-42) questionnaire [32].

*2.5. Statistical Analysis*

Participant data were de-identified before analysis. Cross-sectional comparisons were made for all outcome variables using a paired *t*-test, and effect sizes are presented as Cohen's *d* [33]. Statistical significance was set at $p \leq 0.05$. Statistical tests and graphs were produced using GraphPad Prism version 9.4.1 for Windows, GraphPad Software, San Diego, CA USA, www.graphpad.com.

## 3. Results

*3.1. Implementation*

3.1.1. Reach

In total, 92 IBC were approached about the study, and 79 (86%) consented to participate in the 12-week exercise intervention. Most (*n* = 76) IBC participants were referred to the study by a medical professional (oncologist (*n* = 52); oncology nurse (*n* = 12); family physician (*n* = 10); other allied health care professional (*n* = 2)). The remaining participants (*n* = 16) were self-referred (Figure 2).

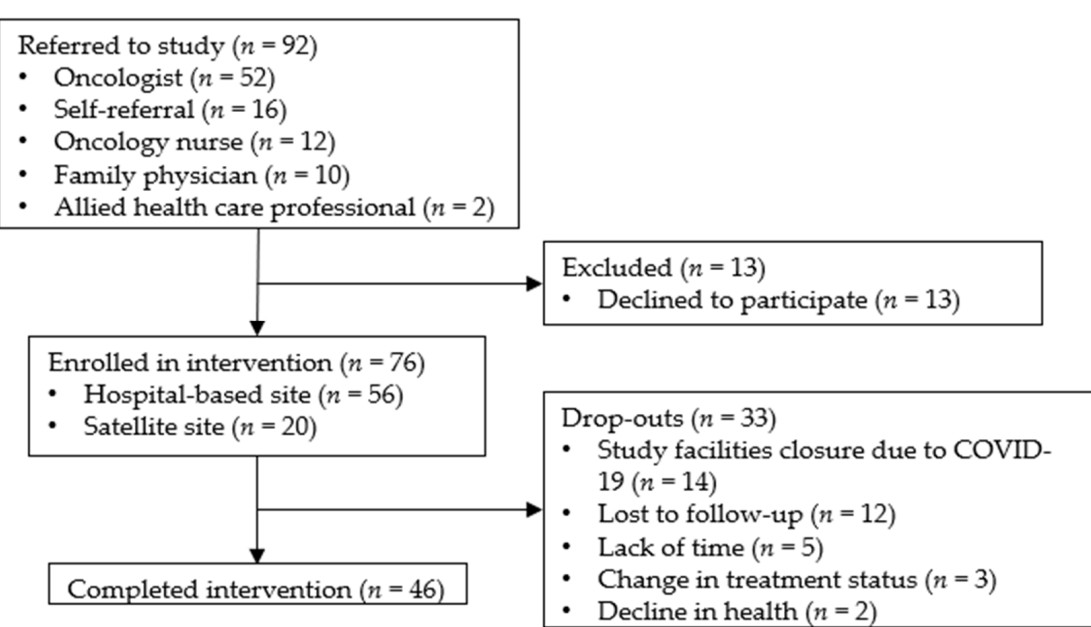

**Figure 2.** Patient disposition for individuals living with and beyond breast cancer referred to the Activating Cancer Communities through an Exercise Strategy for Survivors (ACCESS) Study.

Of the consenting participants, 43 completed the 12-week exercise program, while 36 did not. The primary reason participants could not complete the study was the closure of the fitness facilities because of the COVID-19 pandemic in March 2020, causing the study to shut down while 14 participants were enrolled. COVID-19 dropouts accounted for 39% of the participants who did not complete the program (Figure 2).

Unrelated to COVID-19, 12 participants were lost to follow-up, 5 reported a lack of time, 3 experienced a change in their treatment status, and 2 experienced a decline in health. Such numbers represented 61% of the dropouts. Baseline and post-intervention characteristics and outcome scores for all 43 participants that completed the study are listed in Table 1. Missing values are due to data collection errors in satellite sites staffed with non-research personnel. Given the later addition of the community-based sites (January 2019 and March 2019), most participants were enrolled in the hospital-based program (*n* = 56) (Figure 2).

**Table 1.** Pre- and post-intervention participants' average scores and standard deviation.

| Variable | n | Pre: x̄ ± SD (Range) | Post: x̄ ± SD (Range) | Effect Size |
|---|---|---|---|---|
| Age (years) | 25 | 57.3 ± 10.1 | NA | NA |
| Height (cm) | 23 | 162.1 ± 6.8 | NA | NA |
| 6-MWT (m) | 22 | 491.8 ± 66.5 | 547.3 ± 75.4 * | 1.59 |
| rHR (bpm) | 22 | 78 ± 12 | 81 ± 12 | 0.30 |
| SBP (mmHg) | 22 | 120 ± 17 | 121 ± 19 | 0.01 |
| DBP (mmHg) | 22 | 80 ± 11 | 81 ± 13 | 0.01 |
| Weight (kg) | 23 | 71.2 ± 13.8 | 71.1 ± 14.1 | −0.06 |
| BMI (kg/m$^2$) | 20 | 27.4 ± 5.1 | 27.0 ± 5.4 | −0.20 |
| Waist Circumference (cm) | 23 | 92.2 ± 13.2 | 89.7 ± 12.6 * | −0.67 |
| Hip Circumference (cm) | 23 | 104.9 ± 10.7 | 103.0 ± 9.7 * | −0.65 |
| Hand Grip (R and L) (kg) | 23 | 50.1 ± 7.1 | 49.6 ± 7.6 | −0.04 |
| Balance (R) (s) | 23 | 38.0 ± 14.6 | 37.1 ± 15.4 | −0.16 |
| Balance (L) (s) | 23 | 38.4 ± 13.7 | 38.7 ± 14.3 | 0.02 |
| Balance (Closed, R) (s) | 20 | 11.9 ± 13.3 | 10.1 ± 10.8 | −0.20 |
| Balance (Closed, L) (s) | 20 | 12.6 ± 13.8 | 13.5 ± 14.4 | 0.08 |
| Shoulder (R) (degrees) | 23 | −6.1 ± 7.9 | −5.0 ± 7.2 | 0.31 |
| Shoulder (L) (degrees) | 22 | −9.1 ± 8.1 | −7.9 ± 8.0 | 0.33 |
| Chair Stand | 23 | 14 ± 4 | 16 ± 5 * | 0.90 |
| Sit and Reach (cm) | 23 | 4.0 ± 12.1 | 7.2 ± 8.7 | 0.42 |
| Mild PA/week (min) | 29 | 105 ± 100 (0–420) | 101 ± 93 (0–420) | −0.04 |
| Moderate PA/week (min) | 33 | 91 ± 123 (0–605) | 161 ± 168 * (0–630) | 0.63 |
| Vigorous PA/week (min) | 29 | 14 ± 31 (0–135) | 55 ± 70 * (0–240) | 0.65 |
| MVPA/week (min) | 28 | 91 ± 98 (0–605) | 200 ± 180 * (0–780) | 0.76 |
| LSI | 28 | 26 ± 24 (0–131) | 36 ± 21 * (0–82) | 0.78 |
| Physical Wellbeing | 36 | 23 ± 5 (4–28) | 24 ± 4 * (12–28) | 0.30 |
| Social Wellbeing | 36 | 21 ± 6 (0–28) | 22 ± 5 (11–28) | 0.14 |
| Emotional Wellbeing | 36 | 18 ± 3 (7–24) | 19 ± 4 (10–24) | 0.21 |
| Functional Wellbeing | 36 | 19 ± 7 (0–28) | 20 ± 6 (0–28) | 0.40 |
| Fatigue | 34 | 35 ± 11 (9–52) | 40 ± 9 * (22–52) | 0.52 |
| TOI | 34 | 100 ± 14 (46–124) | 102 ± 12 * (70–122) | 0.12 |
| Stress | 36 | 7 ± 6 (0–34) | 5 ± 6 (0–24) | −0.32 |
| Anxiety | 32 | 3 ± 2 (0–22) | 2 ± 3 (0–10) | −0.23 |
| Depression | 35 | 5 ± 5 (0–37) | 3 ± 3 * (0–10) | −0.43 |
| DASS-42 score | 32 | 13 ± 10 (0–87) | 10 ± 10 (0–37) | −0.40 |
| PSQI score | 26 | 13 ± 3 (8–19) | 12 ± 3 (7–18) | −0.31 |

* Denotes a significant ($p < 0.05$) change from pre to post. Abbreviations: bpm, beats per minute; cm, centimetres; DASS-42, Depression Anxiety Stress Scale; DBP, diastolic blood pressure; FACIT-F, functional assessment of chronic illness-fatigue; FACT-G, functional assessment of cancer therapy-general; GLTEQ, Godin leisure-time exercise questionnaire; kg, kilograms; L, left; LSI, leisure score index; mmHg, millimetres of mercury; MVPA, moderate to vigorous physical activity; MWT, minute walk test; NA, not applicable; PA, physical activity; PSQI, Pittsburgh sleep quality index; R, right; rHR, resting heart rate; SBP, systolic blood pressure; sec, seconds; yrs, years; SD, standard deviation; TOI, trial outcome index.

Exercise program attendance for those who completed the study was 97.9%, and only one adverse event was reported. The adverse event stated that a participant developed a sore knee, and exercises were adapted to the individual's ability.

### 3.1.2. Participant Description

Most consenting participants (*n* = 38) reported receiving surgery, radiation, and chemotherapy, while others received surgery and radiation (*n* = 12), surgery (*n* = 7), surgery and chemotherapy (*n* = 6), radiation and chemotherapy (*n* = 1), and chemotherapy (*n* = 1) only. Three participants experienced metastasis (spine = 1, bone = 1, bone and liver = 1). The time between initial cancer treatment and exercise program initiation ranged from 122 to 589 days, with a median count of 241.

### 3.2. Effectiveness

### 3.2.1. Physical Fitness and Body Composition

The fitness tests were administered before and after the intervention to investigate the impact of a tailored exercise intervention on physical fitness. Distances of the 6-MWT increased significantly from baseline, as did the number of chair sit-to-stand repetitions (Figure 3A,B; Table 1). These changes represent a large effect size. On the other hand, grip strength remained unchanged for both hands (Figure 3C,D). Likewise, there were no significant changes in flexibility outcomes, including shoulder flexibility and the sit and reach scores. However, the sit and reach scores improved for most IBC, but the difference was not statistically significant. Waist and hip circumference significantly decreased from baseline (Table 1). However, resting heart rate, systolic and diastolic blood pressure, weight, and body mass index remained unchanged.

### 3.2.2. Patient-Reported Outcomes

To investigate the impact of a tailored exercise intervention on physical activity behaviour, participants completed the GLTEQ. Following the exercise intervention, IBC had significantly higher levels of moderate- and vigorous-intensity physical activity and tended to have lower levels of mild-intensity physical activity per week, as determined by the GLTEQ (Figure 3E; Table 1). This transition from low to moderate activity led to a significant increase in moderate to vigorous physical activity per week and improved the overall LSI. It is also worth noting that IBC completed more vigorous activity per week following the exercise intervention, but this change was not significant.

To investigate the impact of a tailored exercised intervention on emotional state, participants completed the DASS-42 questionnaire. The DASS-42 questionnaire investigated the levels of depression, anxiety, and stress. Scores on the depression subscale of the DASS-42 improved following the exercise intervention (Figure 4A), while no significant changes in the anxiety or stress subscales were observed. Negative emotional states, as measured by the total DASS-42 score, improved after the exercise intervention (Figure 4D; Table 1).

Participants completed the FACT-G and FACIT-F questionnaires to investigate the impact of a tailored exercised intervention on HRQOL. Overall HRQOL scores improved after the intervention, as measured by the trial outcome index (Figure 5; Table 1). When the FACT-G was broken down into its subscales, it revealed that physical wellbeing scores improved significantly, and the social, emotional, and functional wellbeing subscales showed positive trends in improvement. These improvements in physical wellbeing represented a small effect size. Chronic illness-related fatigue measured using the FACIT-F also significantly improved following the intervention and represented a moderate effect size. Sleep quality remained unchanged as determined by the PSQI (Table 1).

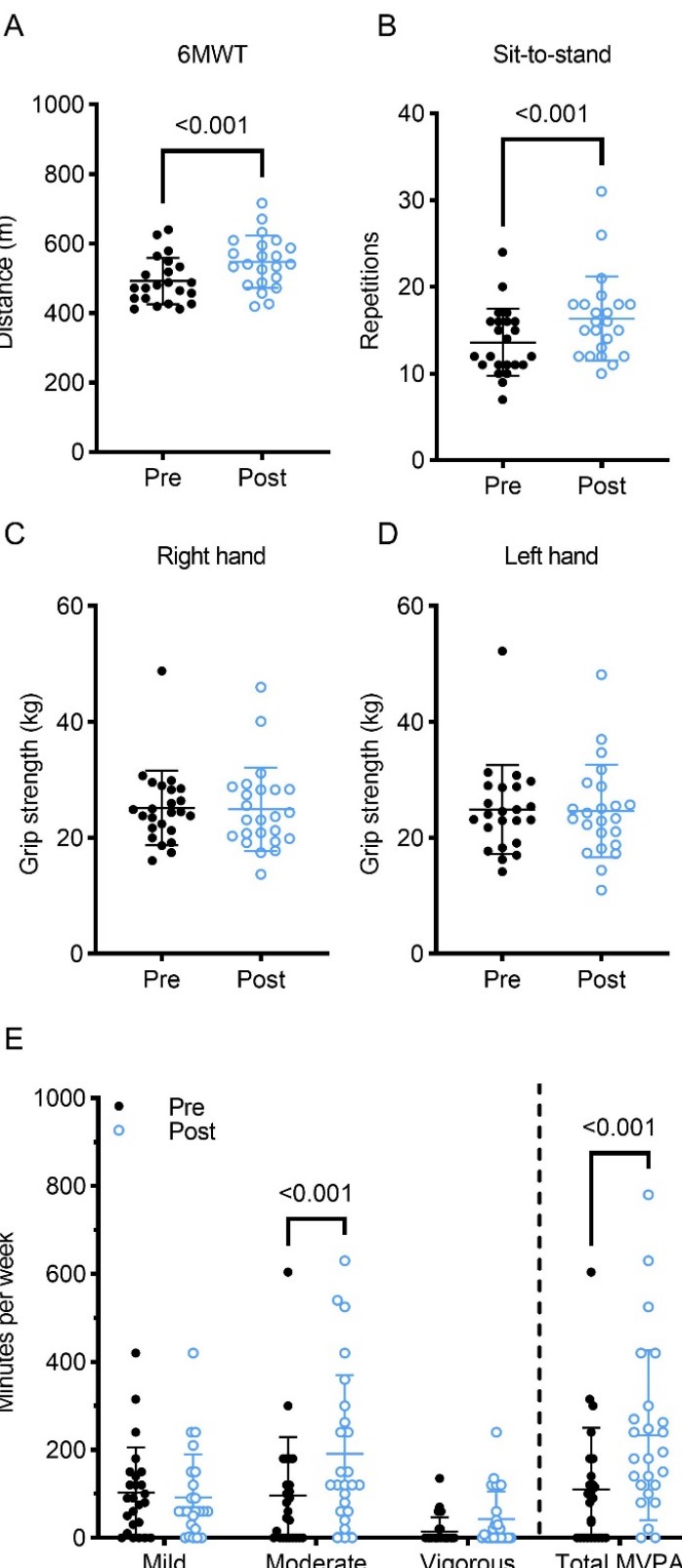

**Figure 3.** (**A**) Change in 6-Minute Walk Test (6MWT) score; (**B**) change in sit-to-stand score; (**C**) change in right-hand grip strength; (**D**) change in left-hand grip strength; (**E**) change in mild, moderate, vigorous, and combined moderate to vigorous physical activity (MVPA) per week.

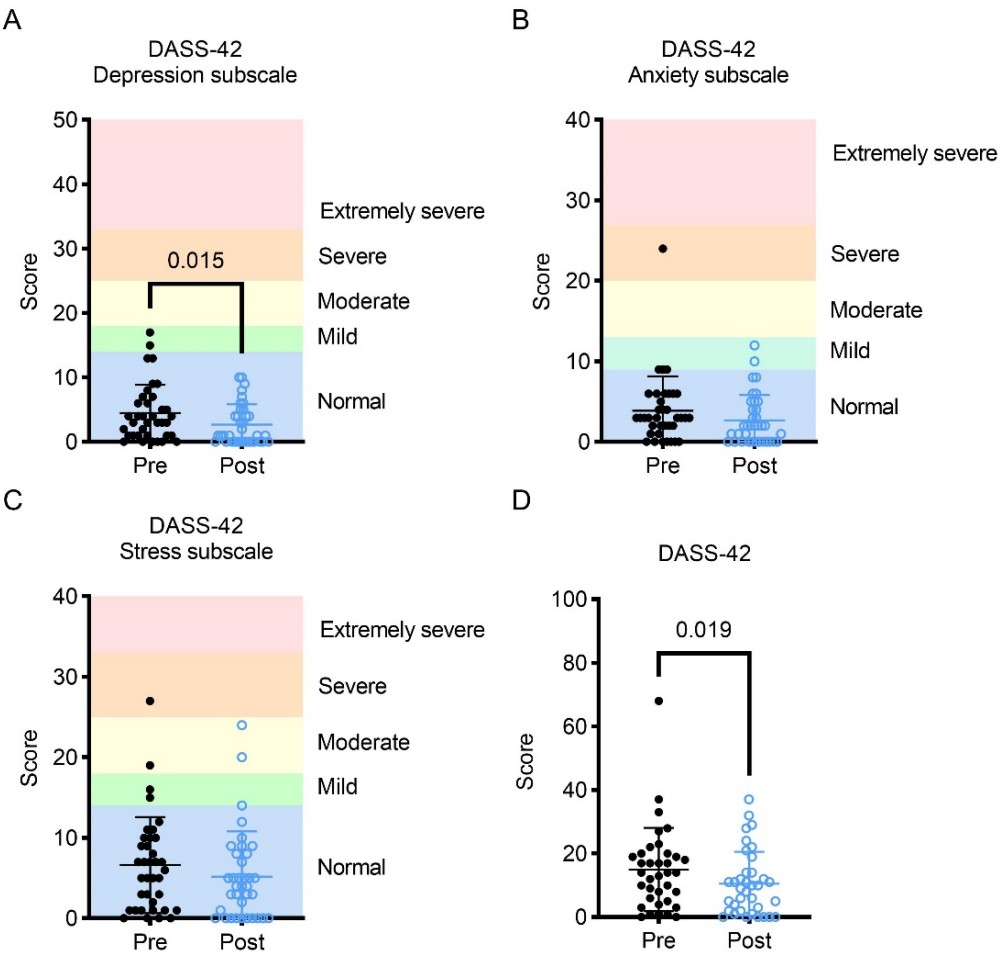

**Figure 4.** (**A**) Change in DASS-42 (Depression, Anxiety and Stress Scale-42 Questions) depression sub-score; (**B**) change in DASS-42 anxiety sub-score; (**C**) change in DASS-42 stress sub-score; (**D**) change in DASS-42 total subscale. Ranges for each subscale were obtained from the DASS manual [32].

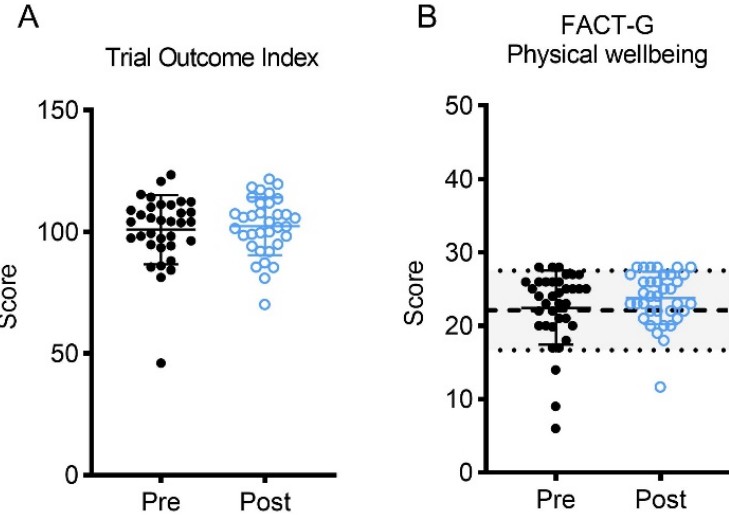

**Figure 5.** *Cont.*

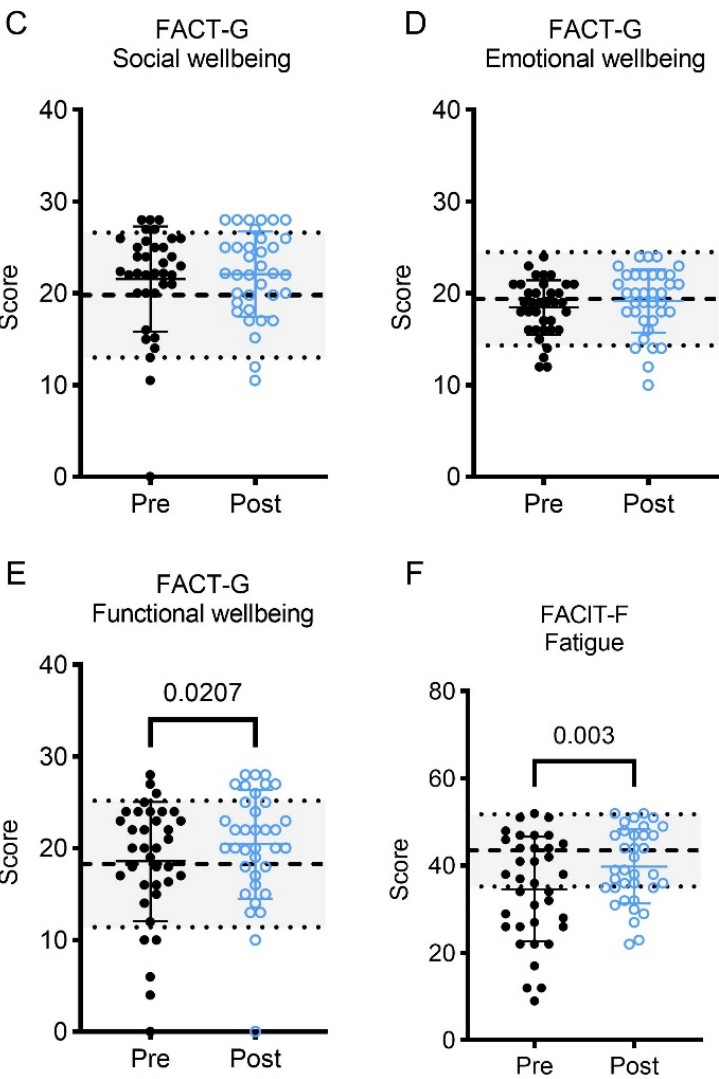

**Figure 5.** (**A**) Change in FACT-G (Functional Assessment of Cancer Therapy-General) Trial Outcome Index score; (**B**) change in FACT-G physical wellbeing subscale; (**C**) change in FACT-G social wellbeing subscale; (**D**) change in FACT-G emotional wellbeing subscale; (**E**) change in FACT-G functional wellbeing subscale; (**F**) change in FACIT-F (Functional Assessment of Chronic Illness Therapy-Fatigue) subscale. Shaded sections reflect the mean ± standard deviation for normative data of American adults for FACT-G [34], and normative values for the FACIT-F are found in Montan et al. [35].

## 4. Discussion

This study demonstrated that a tailored exercise intervention given to IBC in a real-world setting improved aerobic fitness, body composition, lower limb endurance, physical activity behaviour, depression status, physical wellbeing, and fatigue status. Similar to earlier work, this study demonstrated that exercise improves aerobic fitness [36], body composition [37], lower limb muscular endurance [38], physical activity behaviour [39], and decreases IBC fatigue [2,5,6] and depression [40]. This study suggests that similar benefits from exercise can be attained in a real-world setting.

The current study's intensities and exercise program durations were effective and based on participant ability and preference. IBC completed the program in three settings across Nova Scotia. Tailoring exercise prescriptions to each individual's ability and preference is necessary to maintain exercise program adherence and prevent attrition [41]. In the current study, considerations for personal preference (e.g., tailoring the exercises to each participant's needs) were accompanied by improvements in health-related measures. Thus, this study suggests that individually tailoring exercise programs in a real-world setting

improved multiple facets of health in IBC while offering a flexible program that can foster high adherence.

The magnitude of the beneficial effects of exercise for IBC in the current study was often clinically relevant. Notably, the improvements in 6-MWT distance averaged 55.5 m, indicative of clinically meaningful improvements in aerobic fitness [41]. Likewise, lower limb muscular endurance scores from the chair sit-to-stand test increased from 14 to 16 counts indicating that the exercise program changed IBC's physical independence status from "non-independent" to "independent" [42]. Furthermore, the improvements in waist circumference were >2 cm, indicating a clinically relevant change [43]. The current study's change in waist circumference suggests IBC experienced a possible reduction in visceral adiposity, associated with a reduced risk for comorbidities such as cardiovascular [44] and cardiometabolic diseases [45]. The improvements in these physical measures and their clinical relevance highlight the value of implementing exercise programming as standard care for IBC, particularly in a population with a high rate of cardiovascular disease incidence and mortality [44,46].

The effects of the current exercise program on physical fitness yielded comparable results to more controlled studies (i.e., randomized controlled trials). Cornette et al. investigated the impact of a 27-week home-based exercise program on physical fitness, strength, fatigue, quality of life, physical activity levels, and anxiety/depression in women receiving treatment for breast cancer [47]. The study randomly allocated 44 women into a control (*n* = 22) or a combination of aerobic and resistance exercise (*n* = 20) group. On average, exercisers walked 4.68% further during the 6-MWT, but experienced no significant changes in their fatigue, anxiety/depression, or quality of life after completing the program. In the current study, participants walked 10.14% further, on average, in the 6-MWT, the program was successful in multiple locations, and IBC also experienced significant improvements in patient-reported outcomes (i.e., fatigue, physical wellbeing, depression). Similarly, Ibrahim et al. [48] investigated the effects of a 12-week exercise program on shoulder mobility and grip strength in IBC throughout treatment. The group reported that shoulder flexibility tended to be higher, and hand grip strength was maintained in the exercise group, although the findings were not significant. Comparatively, hand grip strength and shoulder flexibility did not significantly change in the current study, but tended to improve. However, the current study did not observe significant improvements in resting heart rate, systolic and diastolic blood pressure, weight, or body mass index. These findings were unsurprising, considering the exercise program did not include specific modifications to promote weight loss (e.g., dietary changes) or improve hemodynamic measures [49]. Overall, employing a tailored, multimodal exercise intervention approach, as conducted in the current study, has the potential to improve multiple aspects of physical and mental health of IBC [50].

Completing this exercise program also improved HRQOL outcomes, such as depression status and general wellbeing (i.e., trial outcome index), which strongly predict a IBC's ability to maintain lifestyle and independence [4]. Notably, after completing the exercise program, IBC scores tended to increase such that most participants scored in the regular or better range for HRQOL measures as indicated by normative data [34] for the trial outcome index, physical wellbeing and fatigue [34,35]. These findings align with the current literature reporting the benefits of exercise on HRQOL [2,4–6], depression [40], and fatigue [51]. Thus, under real-world conditions, a tailored exercise program for IBC induces similar benefits as found in more rigorous and clinical randomized controlled trials.

### 4.1. Limitations

While the current report suggests that a real-world tailored exercise program benefits IBC, the present study has some limitations. Firstly, the current study only assessed IBC health at two time points, pre-and post-intervention. Therefore, a long-term follow-up study is necessary to investigate if IBC adopt an active mindset and continue exercising, as well as elucidate the long-term benefits. Additionally, the duration of the exercise

intervention was limited to 12 weeks. Extending the exercise duration would likely lead to more considerable benefits [40,52,53]. Moreover, incorporating a comparison group receiving a different exercise prescription (i.e., non-tailored) employed in a real-world setting would allow for a more objective analysis of the specific benefits of tailored exercise programming for IBC. Finally, it should be noted that the present study lacks a complete dataset for all participants. Implementing fidelity checks to ensure that satellite sites comply with study protocols would enhance the completeness of the data [54]. These limitations are partly a result of the constrained funding available for the pilot study.

### 4.2. Future Directions

The current sample primarily represented an urban population (Halifax Regional Municipality and Truro (town), NS). Future research engaging rural areas, such as the ongoing Exercise for Cancer to Enhance Living Well (EXCEL; NCT04478851) [54], would allow exercise programs for IBC to have a greater reach. To deliver tailored exercise programs to IBC living in rural areas, researchers must collaborate with community health clinics and recreation centres and provide programs through online platforms to reach IBC without reliable access to transportation. Ensuring all IBC have access to exercise programming will improve health and patient-reported outcomes. Given these improvements, incorporating CEP-guided exercise training into clinical practice would likely prove fruitful for breast cancer care programs; exercise training should be supported by health authorities and implemented using CEPs and other qualified exercise professionals with cancer-specific training. Furthermore, sampling from a larger geographic area will also allow for increased generalizability of research findings and multiple comparisons investigating the implementation effectiveness of the exercise program across socioeconomic groups. Lastly, the high degree of data loss from conducting this implementation–effectiveness study at the satellite sites is noteworthy. It is essential for the success of future trials to ensure that staff at non-research sites can accurately and reliably record data. Defining and overcoming barriers to this will be crucial for the widespread implementation and critical analysis of exercise trials for IBC.

### 5. Conclusions

The current research results suggest that a 12-week individualized exercise program in a real-world setting improves patient-reported outcomes, body composition, physical activity behaviour, and overall fitness and HRQOL in IBC. Reaching IBC in their communities and designing programs around their medical needs and preferences is essential to ensure program adherence and effectiveness. Considering that 12% of all Canadian women will be diagnosed with BC at some point in their lifetime [55], implementing exercise programming as a standard of care should be strongly considered.

**Author Contributions:** Conceptualization, C.B., N.C.-R., M.L.M., M.R.K. and S.A.G.; methodology, N.C.-R., M.L.M., M.R.K. and S.A.G.; investigation, J.C.C., M.R.K., J.M. and S.A.G.; resources, C.B.; data curation, S.J.K. and S.H.; writing—original draft preparation, S.J.K. and S.H.; writing—review and editing, S.J.K., S.H., C.B., J.C.C., J.M., N.C.-R., M.L.M., M.R.K. and S.A.G.; visualization, S.H.; supervision, M.R.K. and S.A.G.; project administration, J.C.C.; funding acquisition, M.R.K. and S.A.G. All authors have read and agreed to the published version of the manuscript.

**Funding:** This research was funded by the QEII Foundation (893206), and the Quebec Breast Cancer Foundation (201700512C18).

**Institutional Review Board Statement:** This study was conducted in accordance with the Declaration of Helsinki and approved by the Research Ethics Board of Nova Scotia Health (File No. 1023682, approved 2 August 2018) for studies involving humans.

**Informed Consent Statement:** Informed consent was obtained from all subjects involved in the study. All analyses are de-identified, and there is no identifying information in the manuscript.

**Data Availability Statement:** The data presented in this study are available on request from the corresponding author.

**Acknowledgments:** We would like to thank our clinician champions, Charlene Robson and Robyn McFarlane, as well as the research staff, Jodi Langley and Deborah Wright, for their dedicated involvement with ACCESS.

**Conflicts of Interest:** The authors declare no conflict of interest.

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
