# Peer review of "Exercise Programming Modelling a Standard of Care Approach Improves Physical Health and Patient-Reported Outcomes in Individuals Living with Breast Cancer: A Pilot Study"

_curroncol, doi:10.3390/curroncol30080522_

Round 1

Reviewer 1 Report

Comments and Suggestions for Authors

The manuscript highlights the significance of physical activity for cancer survivors, a topic that is frequently overlooked despite substantial evidence supporting its positive impact on overall health.

The hypothesis and experimental design are well described. The objective to improve the access to the cancer-specific exercise programming, lacking in a certain territory, is very plausible.

The manuscript is well structured with appropriate sections, clear materials and methods part, comprehensive results description, and consistent conclusions.

The ethical statements are adequate.

The literature is up-to date.

One notable drawback is the relatively small number of participants who completed the study; however, the findings provide a foundation for future research involving larger cohorts.

I would like to display a few comments for your consideration:

- Line 78. "those prescribing exercise must have the training and experience to implement evidence-based recommendations" - the phrase is not very clear. Maybe "must have had" may sound better?

- Line 113: All exercise professionals completed cancer and exercise training for fitness professionals. - could you please specify the training program and the certification?

- Line 135: 2.4.2. Physical Fitness Measures.

this paragraph describes scoring for the shoulder mobility; I assume following this logic all scoring systems should be indicated or totally removed.

- Line 217:  Three participants experienced metastasis (spine = 1, bone = 2, liver = 1). - was there one patient with two metastasis? please, specify.

- Additionally, in "Patricipants' description" (3.1.2.), might be important to indicate the number of patients with axillary web syndrome and lymphedema (as many of them have undergone surgery), and if this affected the patients' performance or their ability to conclude the study program. 

- Line 240: "resting heart rate, systolic and diastolic blood pressure, weight, and body mass index remained unchanged" - I think this part deserves a discussion.

Thank You.

Author Response

Point 1- Line 78. "those prescribing exercise must have the training and experience to implement evidence-based recommendations" - the phrase is not very clear. Maybe "must have had" may sound better?

Response 1- Thank you for your suggestion, line 78 has been modified accordingly

Point 2- Line 113: All exercise professionals completed cancer and exercise training for fitness professionals. - could you please specify the training program and the certification?

Response 2- The training program is now specified in lines 114-115

Point 3- Line 135: 2.4.2. Physical Fitness Measures.

this paragraph describes scoring for the shoulder mobility; I assume following this logic all scoring systems should be indicated or totally removed.

Response 3- Shoulder mobility scoring description was removed and the change is evident in line 155

Point 4- Line 217:  Three participants experienced metastasis (spine = 1, bone = 2, liver = 1). - was there one patient with two metastasis? please, specify.

Response 4- The description was clarified. 1 participant experienced metastases to the spine, 1 to the bone, and 1 to the bone and liver. The change is evident in line 218 to 219.

Point 5- Additionally, in "Patricipants' description" (3.1.2.), might be important to indicate the number of patients with axillary web syndrome and lymphedema (as many of them have undergone surgery), and if this affected the patients' performance or their ability to conclude the study program. 

Response 5- I agree this information would be interesting to include, especially given the past hesitancy of encouraging individuals with lymphedema to complete resistance training. However, we did not specifically gather information on participant axillary web syndrome or lymphedema. That said, no participants dropped out of the program due to lymphedema exacerbation. Dropout reasons are provided in lines 195 to 202.

Point 6- Line 240: "resting heart rate, systolic and diastolic blood pressure, weight, and body mass index remained unchanged" - I think this part deserves a discussion.

Response 6- Thank you for noticing this. An additional description has been added to the discussion in lines 330 to 336.

Reviewer 2 Report

Comments and Suggestions for Authors

The subject of this manuscript concern the results of a 12-week individualized exercise program, within the study Activating Cancer Communities through an Exercise Strategy for Survivors. This work may be important for IBC patients who do not believe in the very importance of daily exercise.

To the authors:

*Materials and methods: The project has been written clearly, the conditions of inclusion and exclusion are very visible.

*Exercise Intervention: Additional photo insertion will be clearer.

*A very important drawback is the lack of a control group that would allow for an objective comparison with other groups or other interventions.

*Results:

Please, insert the patient recruitment scheme, it will be more readable.

Author Response

Point 1- *Exercise Intervention: Additional photo insertion will be clearer.

Response 1- Thank you for this suggestion. However, the participants in the current study did not complete a media consent, and thus, we cannot include photographs of the participants completing exercises.

Point 2- *A very important drawback is the lack of a control group that would allow for an objective comparison with other groups or other interventions.

Response 2- Thank you for this suggestion. The current study was an implementation-effectiveness study in a real-world setting. However, we agree that a control group would improve the objective interpretation of the analysis. Accordingly, a section has been added to directly address this limitation in lines 353 to 356.

Point 3- Results* Please, insert the patient recruitment scheme, it will be more readable.

Response 3- I appreciate this idea. A CONSORT diagram (patient disposition figure) has been added in lines 211-213

Reviewer 3 Report

Comments and Suggestions for Authors

The authors have produced an interesting and useful study to evaluate how to implement an individualized  exercise intervention delivered to individuals living with breast cancer, the effects of the exercise intervention on individuals living with breast cancer  physical health and patient-reported outcomes in a real-world setting.

However, I would like to make some observations before recommending your work for publication.

1. Could the authors add the type of study in the title?

2. Could the authors provide a graph of the intervention performed?

3. In the introduction/discussion section, I recommend the authors comment on exercise interventions in other types of populations: doi:10.1093/pm/pnz036

4. Could the authors provide a graphical abstract of the study? 

5. In the Discussion section, could you add a section on "Clinical Implications"?

6. Finally, although it is not indispensable, I suggest the authors to provide some images of the intervention.

Author Response

The authors have produced an interesting and useful study to evaluate how to implement an individualized exercise intervention delivered to individuals living with breast cancer, the effects of the exercise intervention on individuals living with breast cancer. physical health and patient-reported outcomes in a real-world setting.

However, I would like to make some observations before recommending your work for publication.

  1. Could the authors add the type of study in the title?

Thank you for this feedback, the change is reflected in line 4

2. Could the authors provide a graph of the intervention performed?

We appreciate this point. A graph of the intervention is provided in lines 124-128

3. In the introduction/discussion section, I recommend the authors comment on exercise interventions in other types of populations: doi:10.1093/pm/pnz036

Thank you for this suggestion, changes have been made in lines 38-40

  1. Could the authors provide a graphical abstract of the study? 

This has been added in line 16

  1. In the Discussion section, could you add a section on "Clinical Implications"?

Clinical implications added in lines 373-377

  1. Finally, although it is not indispensable, I suggest the authors to provide some images of the intervention.

We appreciate this thought. However, this is not possible as the participants did not consent to photographs, and no images were taken of the exercise intervention.

Round 2

Reviewer 1 Report

Comments and Suggestions for Authors

Thank you for addressing the comments. All the points are now clear to me. 

Reviewer 2 Report

Comments and Suggestions for Authors

The authors have sufficiently improved the manuscript.

In this form, the article can be published.

Reviewer 3 Report

Comments and Suggestions for Authors

Congratulations to the authors for the final version of their manuscript.